# Deconstructing and repurposing the light-regulated interplay between *Arabidopsis* phytochromes and interacting factors

David Golonka[1], Patrick Fischbach[2], Siddhartha G. Jena[3], Julius R.W. Kleeberg[1], Lars-Oliver Essen[4], Jared E. Toettcher[3], Matias D. Zurbriggen[2]* & Andreas Möglich[1,5,6,7]*

Phytochrome photoreceptors mediate adaptive responses of plants to red and far-red light. These responses generally entail light-regulated association between phytochromes and other proteins, among them the phytochrome-interacting factors (PIF). The interaction with *Arabidopsis thaliana* phytochrome B (*At*PhyB) localizes to the bipartite APB motif of the *A. thaliana* PIFs (*At*PIF). To address a dearth of quantitative interaction data, we construct and analyze numerous *At*PIF3/6 variants. Red-light-activated binding is predominantly mediated by the APB N-terminus, whereas the C-terminus modulates binding and underlies the differential affinity of *At*PIF3 and *At*PIF6. We identify *At*PIF variants of reduced size, monomeric or homodimeric state, and with *At*PhyB affinities between 10 and 700 nM. Optogenetically deployed in mammalian cells, the *At*PIF variants drive light-regulated gene expression and membrane recruitment, in certain cases reducing basal activity and enhancing regulatory response. Moreover, our results provide hitherto unavailable quantitative insight into the *At*PhyB:*At*PIF interaction underpinning vital light-dependent responses in plants.

[1] Lehrstuhl für Biochemie, Universität Bayreuth, 95447 Bayreuth, Germany. [2] Institute of Synthetic Biology and CEPLAS, Heinrich Heine University Düsseldorf, 40225 Düsseldorf, Germany. [3] Department of Molecular Biology, Princeton University, Princeton, NJ 08544, USA. [4] Department of Chemistry, Center for Synthetic Microbiology, Philipps University Marburg, 35032 Marburg, Germany. [5] Research Center for Bio-Macromolecules, Universität Bayreuth, 95447 Bayreuth, Germany. [6] Bayreuth Center for Biochemistry & Molecular Biology, Universität Bayreuth, 95447 Bayreuth, Germany. [7] North-Bavarian NMR Center, Universität Bayreuth, 95447 Bayreuth, Germany. *email: matias.zurbriggen@uni-duesseldorf.de; andreas.moeglich@uni-bayreuth.de

First discovered among the plant photoreceptors[1], phytochromes (Phy) sense red and far-red light to control a range of physiological responses, including seedling germination, shade avoidance, entrainment of the circadian clock, and the transition from vegetative to reproductive growth[2]. Beyond plants, Phys also occur in bacteria and fungi where they mediate chromatic adaptation and pigmentation among other processes[3,4]. Receptors of the Phy family generally exhibit a bipartite architecture with an N-terminal photosensory core module (PCM) and a C-terminal output module (OPM) (Fig. 1a). The PCM of canonical Phys comprises consecutive PAS (Per/ARNT/Sim), GAF (cGMP-specific phosphodiesterase, adenylyl cyclase, and FhlA), and PHY (Phy-specific) domains and binds within its GAF domain a linear tetrapyrrole (bilin) chromophore[3,5] (Fig. 1b). Phys of higher plants naturally employ phytochromobilin (PΦB), covalently attached to a cysteine residue within the GAF domain, but can be functionally reconstituted with phycocyanobilin (PCB) of cyanobacterial origin. In darkness, conventional Phys adopt their red-absorbing Pr state with the bilin chromophore in the 15Z configuration; absorption of red light triggers rapid bilin isomerization to the 15E state and population of the metastable, far-red-absorbing Pfr state (Fig. 1b). The Pfr → Pr reversion occurs thermally or can be actively driven by far-red light. Insight from bacterial Phys illustrates that the Z/E isomerization is coupled to refolding of the so-called PHY tongue, a protrusion of the PHY domain, from a β-hairpin to an α-helix conformation, in turn prompting quaternary structural rearrangements[6–10]. Bacterial Phys mostly form part of two-component signaling cascades[11] with OPMs acting as histidine kinases (HKs). By contrast, the Phy OPMs of land plants comprise two PAS domains, PAS-A and PAS-B, and a homologous HK-related domain that, however, lacks key residues essential for function and is thus devoid of HK activity. Rather, plant Phys have been reported to exhibit serine/threonine kinase activity[12,13]. Plant Phys exert their biological effects via light-regulated cytonucleoplasmic shuttling and protein:protein interactions (PPIs), which manifest in transcriptional responses and proteolytic degradation of cellular target proteins[14–16]. As one prominent protein family, the so-called phytochrome-interacting factors (PIFs) undergo light-regulated PPIs with plant Phys and act as basic helix–loop–helix transcription factors[14,17–20] (Fig. 1c and Supplementary Fig. 1).

*Arabidopsis thaliana* possesses five Phys, denoted AtPhyA–E, that engage with a set of at least eight PIFs, denoted AtPIF1–8[14,17]. For the arguably best-studied Phy, AtPhyB, preferential interactions of the Pfr state vs. the Pr state were identified with all eight AtPIFs[14,17,21–23]. Notably, the PCM of AtPhyB is necessary and sufficient for red-light-activated and far-red-light-reversible AtPIF binding[19,24–26]. Although a pioneering study on the optogenetic use of AtPhyB reported that reversible interactions with AtPIF6 required the presence of PAS-A and PAS-B[27], numerous later studies demonstrated that the PCM suffices for photoreversible interactions with AtPIFs[28–30]. That notwithstanding, the C-terminal OPM likely contributes to light-regulated PPIs and is integral to eliciting physiological responses[14,15]. Likewise, the light-activated interaction with AtPhyB maps to the weakly conserved APB (active phytochrome B binding) consensus motif within the N-terminal region of AtPIF orthologs that precedes the basic helix–loop–helix domain[24] (Fig. 1d). The APB motif consists of two segments, termed APB.A and APB.B, the first of which exhibits higher sequence conservation (Supplementary Fig. 1) and dominates light-activated AtPhyB binding as indicated by site-directed mutagenesis[24]. In the case of AtPhyA, the isolated PCM also suffices for light-regulated interactions with AtPIF1 and AtPIF3, which localize to the APA motifs (active phytochrome A binding) of these PIFs, somewhat C-terminal of the APB motifs[31,32].

Early on, the light-regulated AtPhy:AtPIF PPI has been harnessed for the control of cellular processes in heterologous hosts by red and far-red light[25,27], an approach now known as optogenetics[33]. As manifold natural processes are intrinsically governed by PPIs, the AtPhy:AtPIF system provides a widely

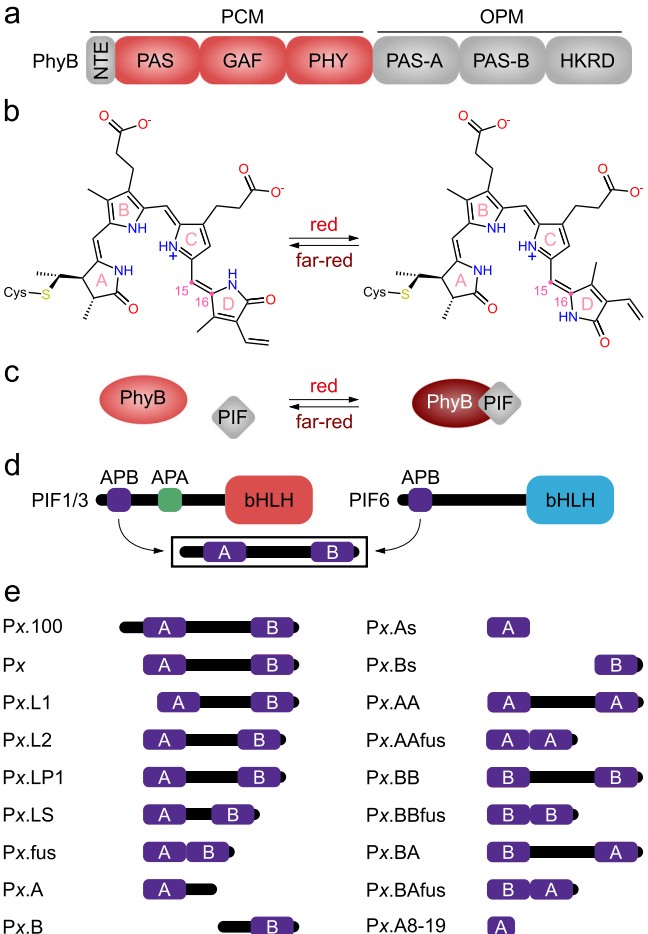

**Fig. 1** Architecture and function of plant phytochromes (Phy) and their cognate phytochrome-interacting factors (PIFs). **a** Modular composition of plant phytochromes. An N-terminal extension (NTE) is succeeded by the photosensory core module (PCM) consisting of consecutive PAS, GAF, and PHY domains, with a phytochromobilin (PΦB) chromophore covalently bound as a thioether within the GAF domain. The C-terminal output module (OPM) comprises two additional PAS domains (PAS-A and PAS-B), succeeded by a histidine-kinase-related domain (HKRD). **b** In the dark-adapted Pr (red-absorbing) state of the Phy, the PΦB chromophore adopts its 15Z form. Red light drives isomerization to the 15E form to give rise to the Pfr state (far-red-absorbing). Vice versa, far-red light drives the Pfr → Pr transition. **c** In their Pr state (red), plant Phys show no or at most weak interactions with PIFs. Following red-light absorption, the Pfr state (brown) is populated and affinity for the PIFs enhanced. **d** Modular composition of PIFs. An N-terminal region of around 100 residues contains the so-called APB motif that mediates interactions with phytochrome B. The APB motif further subdivides into the ABP.A and APB.B segments[24]. Certain PIFs also possess a more C-terminal APA motif engaged in interactions with phytochrome A. The C-terminal part comprises a basic helix–loop–helix (bHLH) DNA-binding domain. **e** Based on the N-terminal fragments of *Arabidopsis thaliana* PIFs 3 and 6, a panel of PIF variants were generated and probed for light-dependent protein:protein interactions with the PCM of *A. thaliana* PhyB (cf. Supplementary Table 1 for a detailed description of these derivatives).

applicable means for the bimodal control of cellular phenomena with supreme resolution in space and time[34]. As a case in point, the expression of transgenes in yeast and mammalian cells has been subjected to red-/far-red-light control via a two-hybrid strategy[25,35,36]. To this end, a split transcription factor was engineered with one component of the *At*Phy:*At*PIF pair connected to a sequence-specific DNA-binding domain and the other to a transcriptional *trans*-activating domain. Exposure to red light prompts colocalization of the two entities and onset of expression from synthetic target promoters. In another approach[27,37,38], the *At*Phy:*At*PIF pair conferred light sensitivity on plasma membrane recruitment and cellular signaling cascades in mammalian cells. Although details differ, optogenetic applications to date mostly employ the isolated PCM of *At*PhyB and the N-terminal 100 amino acids of *At*PIF3/6, denoted P3.100 and P6.100, that comprise the APB motif.

Despite the eminent role of the *At*Phy:*At*PIF interaction in nature and optogenetics, quantitative data on the interaction strength and the underlying sequence determinants are scarce. To fill this gap, we dissected and analyzed the light-dependent interaction between *At*PhyB and *At*PIF3/6 by several qualitative and quantitative approaches. Whereas the *At*PhyB PCM bound P6.100 with about 10 nM affinity in its Pfr state and showed no detectable affinity in the Pr state, P3.100 exhibited weaker Pfr-state affinity and elevated basal affinity in Pr. By deconstructing *At*PIF3/6 and engineering a wide set of shortened variants, we pinpointed APB.A as decisive for light-regulated PPIs, with a modulatory role for APB.B. Quantitative analyses informed the construction of minimal *At*PIF3/6 fragments of 25 and 23 residues, respectively, that retained stringently light-regulated PPIs with *At*PhyB. When deployed for the optogenetic control of gene expression and membrane recruitment, the novel *At*PIF variants with a range of interaction strengths achieved stratified and enhanced light responses.

## Results

**Deconstructing the *At*PhyB:*At*PIF interaction.** Starting from the *At*PIF constructs P3.100 and P6.100, we generated numerous derivatives with residues deleted from the N terminus, the linker between the APB.A and APB.B segments varied, or either segment omitted or duplicated (Fig. 1e, Supplementary Table 1). All *At*PIF variants were C-terminally tagged with enhanced yellow fluorescent protein (EYFP) to promote protein solubility and facilitate concentration determination. We implemented a screening assay to efficiently probe interactions of these variants with the Pfr state of the *At*PhyB PCM. The screen exploits the fact that *At*PIF binding stabilizes the Pfr state of *At*PhyB and decelerates the thermal reversion to the Pr state in the dark[39] (Fig. 2a). For this assay, the *At*PIF-EYFP variants were expressed in *Escherichia coli*, purified *At*PhyB PCM was added to the crude cell lysate in substoichiometric amounts, and the Pfr → Pr reversion kinetics were monitored by absorption spectroscopy (Fig. 2b, c). The initial kinetics were normalized to an EYFP-negative control and provide a convenient readout for interactions (Fig. 2d). Although qualitative in nature, this first screening platform offers important advantages: (i) owing to the specificity of the *At*PhyB:*At*PIF interaction, the assay can be conducted in crude bacterial lysate, without the need for protein purification; and (ii) it can be easily multiplexed to test many variants in a single experiment.

A multiple sequence alignment of the N-terminal regions of *At*PIF1–8 delineates two regions of conservation that define the A and B segments of the APB motif (Supplementary Fig. 1[24]). The APB.A segment shows stronger conservation and comprises

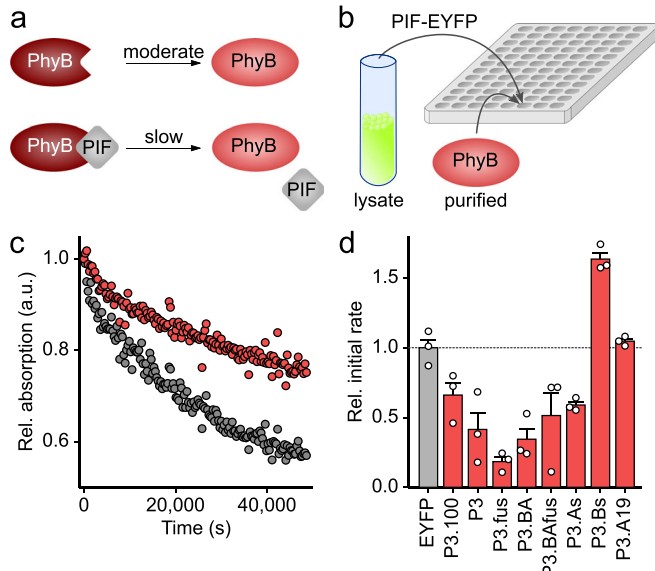

**Fig. 2** Screening *At*PIF variants for protein:protein interactions with the *At*PhyB PCM. **a** The light-adapted Pfr state (brown) of *At*PhyB thermally recovers to the dark-adapted Pr state (red) in a moderately paced reaction. When binding to an *At*PIF variant, the recovery reaction is delayed. **b** *At*PIF variants were C-terminally tagged with EYFP, expressed in *Escherichia coli*, cells were lysed, and *At*PhyB PCM was added to the crude lysate. Samples were exposed to red light, and the recovery reaction was monitored over time by absorption measurements. **c** Normalized absorption of the *At*PhyB PCM measured at 720 nm after red-light absorption in the presence of P3.100 (red) or the EYFP-negative control (gray). **d** The initial rates of the recovery reaction were determined and normalized to the reading obtained for the EYFP-negative control. Data indicate mean ± SEM of $n = 3$ independent biological replicates.

around 20 residues centered around the consensus core sequence ELXXXXGQ[24]; by comparison, the APB.B region is considerably shorter and less conserved. As the very N-terminal region preceding APB.A varies substantially among the *At*PIFs in length and sequence, we deemed it non-essential for *At*PhyB interactions and removed it from P3.100 and P6.100. The resultant P*x* variants (here and in the following, $x = 3, 6$) retained interaction with the *At*PhyB PCM, and all subsequent *At*PIF variants were thus based on these N-terminally truncated forms (Fig. 2d and Supplementary Fig. 2). Next, we interrogated the linkage between the constituent APB.A and APB.B segments, which is of heterogenous length and sequence across *At*PIF1–8. We generated a set of variants, including (i) P*x*.L1 and P*x*.L2 in which the linkers of P3/P6 are shortened by 10 residues at their N and C termini, respectively; (ii) P*x*.LP1 in which said linker is substituted for the corresponding segment of *At*PIF1, the shortest among all *At*PIFs; and (iii) P*x*.LS in which the linker is replaced by a repetitive glycine–serine stretch of 10 residues. As gauged by their effect on dark-reversion kinetics (cf. Supplementary Fig. 2), all these variants still interacted with the Pfr state of the *At*PhyB PCM. These results imply that the linker connecting the APB.A and APB.B segments is dispensable, which is confirmed in the P*x*.fus variants that directly link these two segments without any linker and still exhibit interaction with the *At*PhyB PCM (cf. Fig. 2d and Supplementary Fig. 2). To assess whether productive *At*PhyB binding mandates a specific topology of the APB segments, we generated the variants P*x*.BA and P*x*.BAfus with

inverted sequential order of APB.A and APB.B, and the original linker sequence kept or removed, respectively. Again, these variants retained interactions with the Pfr state of the AtPhyB PCM (cf. Fig. 2d and Supplementary Fig. 2). Site-directed mutagenesis had previously ascribed a dominant role to APB.A in mediating the light-dependent interaction with AtPhyB[24], and we hence probed the two segments of the composite APB motif separately. Both the APB.A-containing variants Px.A and the Px. As, with or without the N-terminal half of the respective linker, still showed interactions with the AtPhyB PCM as judged by the effect on dark-reversion kinetics (cf. Fig. 2d and Supplementary Fig. 2). By contrast, neither the APB.B-based Px.B nor the Px.Bs variants, with or without the C-terminal half of the linker, respectively, exhibited interactions in this assay. Duplication of the A part in the variants Px.AA and Px.AAfus preserved interactions with the AtPhyB PCM, and vice versa, duplication of the B segment in Px.BB and Px.BBfus failed to restore them (cf. Supplementary Fig. 2). Taken together, our findings emphasize the dominant role of APB.A for mediating interactions with AtPhyB. To further characterize the APB.A segment, we successively trimmed residues flanking its ELXXXXGQ core sequence. However, even the removal of five weakly conserved C-terminal residues in the variants Px.A19 abolished interactions with AtPhyB, as judged by their inability to slow down the AtPhyB-PCM Pfr → Pr reversion kinetics (cf. Fig. 2d and Supplementary Fig. 2). Likewise, no interaction with the AtPhyB PCM was detected for more extensive truncations of the APB.A segment (cf. Supplementary Fig. 2).

**Biochemical analyses of the AtPhyB:AtPIF interaction.** The above screening platform affords a qualitative first-pass assessment of the AtPIF variants but does not quantify the strength of interactions with AtPhyB. Moreover, the assay is limited to interactions within the Pfr state but not the Pr state. We hence selected several of the above AtPIF candidates for in-depth analysis. Following expression and purification, we assessed the oligomeric state of these variants and of the AtPhyB PCM by size-exclusion chromatography (SEC). In its Pr state, the isolated AtPhyB PCM elutes as a monomer with a minor homodimeric fraction, consistent with a recent SEC analysis[40] (Fig. 3a). In the Pfr state, the predominantly monomeric state is maintained but the retention from the SEC column is slightly delayed, which arguably reflects light-induced conformational changes, i.e., a compaction, of the PCM that may resemble those observed in bacterial Phys[6–8,10] (Fig. 3b). At a concentration of 10 μM, P3.100 and P6.100 largely eluted as homodimers with a minor monomeric population (Fig. 3c, Supplementary Fig. 3 and Table 1). Dimerization is not caused by the EYFP tag as the fluorescent protein itself eluted as a monomer (Fig. 3d, Table 1). Notably, the homodimeric state of AtPIFs is also observed in nature and critical for their physiological function as basic helix–loop–helix transcription factors[41]. Size reduction of the AtPIFs impaired homodimerization in several variants to different extent (Supplementary Fig. 3, Table 1). If the APB.A segment was truncated, as in P3.A and P6.A, or excluded altogether, as in P3.Bs and P6. Bs, homodimerization was lost completely. Taken together, these findings point toward a contribution of the APB.A segment to homodimerization of the current AtPIF variants and, by extension, of the intact AtPIF3 and AtPIF6 proteins[41].

We next investigated the interactions between the AtPIF3/6 variants and the AtPhyB PCM by SEC (Fig. 3e, Supplementary Fig. 4 and Table 1). To this end, we first converted the AtPhyB PCM to its Pfr state by illumination with red light (640 nm), incubated it at a 5:1 molar ratio with the different AtPIF variants,

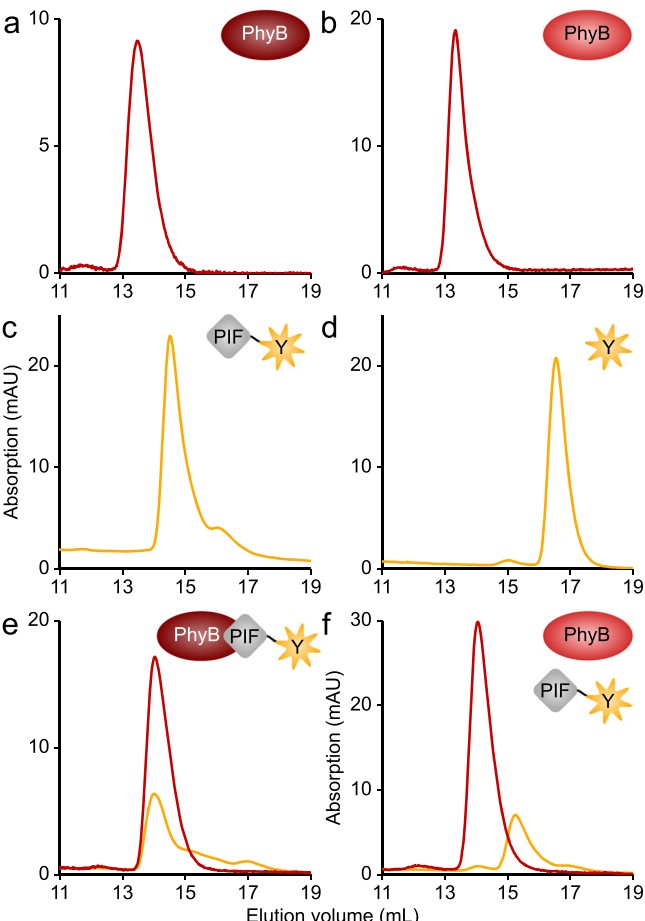

**Fig. 3** Oligomeric state of the AtPIF variants and light-dependent interactions with the AtPhyB PCM. **a** 50 μM AtPhyB PCM were exposed to red light and analyzed by size-exclusion chromatography (SEC), where the yellow and red lines represent absorption at 513 and 650 nm, respectively. **b** As in **a** but the AtPhyB PCM was exposed to far-red light prior to chromatography. **c** 10 μM P3.100-EYFP were analyzed by SEC. Elution profiles were independent of illumination. **d** 10 μM of the negative control EYFP were analyzed by SEC. Elution profiles were independent of light. **e** A mixture of 10 μM P3.100-EYFP and 50 μM AtPhyB PCM was exposed to red light and analyzed by SEC. **f** As in **e** but samples were illuminated with far-red light, rather than red light. Experiments were repeated twice with similar results.

and analyzed the mixture by SEC. In full agreement with the first-pass screening (cf. Fig. 2 and Supplementary Fig. 2), all variants that we had identified as binding-competent exhibited interactions with AtPhyB PCM at an apparent 1:1 stoichiometry. Vice versa, the AtPIF variants that had failed to decelerate AtPhyB reversion kinetics (cf. Fig. 2 and Supplementary Fig. 2) lacked any interactions (Supplementary Fig. 4). We also assessed interactions between the AtPIF variants and the AtPhyB PCM in the Pr state following exposure to far-red light (720 nm) (Fig. 3f and Supplementary Fig. 5). None of the variants showed interactions under these conditions. Insofar red-light-activated binding to the AtPhyB PCM had been retained in the truncated AtPIF variants, far-red light hence abolished it.

Having engineered a suite of AtPIF variants undergoing light-regulated PPIs with the AtPhyB PCM, we next sought to quantify the strength of these interaction in both the Pr and Pfr states.

**Table 1 Biochemical analyses of the *At*PIF3/6 variants.**

| Name | Oligomeric state[a] | *At*PhyB-PCM interaction[a,b] | Pfr state $K_D$ (nM)[c] | Pr state $K_D$ (nM)[c] |
|---|---|---|---|---|
| P3.100 | Homodimer | + | 200 ± 70 | >2000 |
| P6.100 | Homodimer | + | 10 ± 8 | n.d. |
| P3 | Homodimer | + | 220 ± 40 | >10,000 |
| P6 | Homodimer/monomer | + | 10 ± 7 | n.d. |
| P3.fus | Monomer | + | 270 ± 60 | >10,000 |
| P6.fus | Homodimer | + | 200 ± 90 | >10,000 |
| P3.A | Homodimer | + | 220 ± 40 | >10,000 |
| P6.A | Monomer | + | 280 ± 100 | >2000 |
| P3.As | Monomer | + | 680 ± 60 | >10,000 |
| P6.As | Monomer | + | 710 ± 80 | >10,000 |
| P3.AA | Homodimer | + | 370 ± 40 | >3000 |
| P6.AA | Homodimer | + | 360 ± 40 | >2000 |
| P3.AAfus | Homodimer/monomer | + | 230 ± 50 | >10,000 |
| P6.AAfus | Homodimer/monomer | + | 230 ± 30 | >10,000 |
| P3.A19 | Monomer | − | >1000 | n.d. |
| P6.A19 | Monomer | − | >2000 | n.d. |
| P3.A14 | Monomer | − | n.d. | n.d. |
| P6.A14 | Monomer | − | n.d. | n.d. |
| P3.A8 | Monomer | − | n.d. | n.d. |
| P6.A8 | Monomer | − | n.d. | n.d. |
| P3.B | Monomer | − | n.d. | n.d. |
| P6.B | Monomer | − | n.d. | n.d. |
| EYFP | Monomer | − | n.d. | n.d. |

*n.d.* not detectable
[a]As determined by size-exclusion chromatography
[b]A "+" sign indicates that an interaction could be detected by size-exclusion chromatography, a "−" sign denotes that no interaction was observed
[c]As determined by fluorescence anisotropy

Notably, detailed quantitative data of that type are largely unavailable but would tremendously improve our understanding of the *At*PhyB:*At*PIF PPI and inform its optimization. To this end, we resorted to fluorescence anisotropy measurements of the EYFP moiety C-terminally appended to all the *At*PIF variants. Binding of a given *At*PIF-EYFP variant to the *At*PhyB PCM would increase its effective hydrodynamic radius, slow down rotational diffusion, and thus increase fluorescence anisotropy (Fig. 4a). We hence incubated a constant 20 nM of the *At*PIF-EYFP variants with increasing amounts of *At*PhyB PCM under red or far-red light and recorded binding isotherms. The reference construct P6.100 exhibited strong binding to the *At*PhyB PCM under red light but no detectable binding under far-red light even at *At*PhyB-PCM concentrations of 2 µM (Fig. 4b). When calculating dissociation constants ($K_D$), one must consider that red light not only drives the Pr → Pfr transition of Phys but also the reverse Pfr → Pr process. Consequently, continuous illumination with red light (640 nm) leads to population of a photostationary state with a mixed Pfr/Pr population at a ratio of ~0.56/0.44[42] (Fig. 4c). Correcting for the actual fraction in the Pfr state, we determined a $K_D$ for the P6.100:*At*PhyB-PCM pair of 10 ± 8 nM (Table 1). This value is in good agreement with an earlier estimate for this pair of 20-100 nM within mammalian cells based on fluorescence microscopy[27]. In comparison to P6.100, P3.100 exhibited a weaker $K_D$ of 200 ± 70 nM in Pfr and an elevated basal affinity in Pr, with an estimated $K_D$ on the order of low micromolar (Fig. 4d and Table 1). This residual interaction could in principle be due to partial population of the Pfr state of the *At*PhyB PCM under the chosen illumination conditions; however, the absence of basal affinity in case of P6.100 strongly argues against this notion. The slightly weaker affinity and much less pronounced light effect in P3.100 compared to P6.100 may account for the previously reported inability to detect light-regulated interactions of *At*PIF3 with the *At*PhyB PCM in mammalian cells[27]. We then recorded

binding isotherms under red and far-red light for all the *At*PIF variants we had purified and analyzed by SEC (Supplementary Figs. 4 and 5, Table 1). Consistent with our first-pass assessment (cf. Fig. 2d and Supplementary Fig. 2), the removal of the non-conserved N-terminal residues preceding the APB.A segment had no influence on the Pfr interaction. Unexpectedly, omission of these residues in the *At*PIF3 context substantially attenuated the basal Pr-state affinity. For the *At*PIF3 variants, removal of the linker and the APB.B part had no or at most modest effects on affinity to the Pfr state (Supplementary Fig. 6, Table 1). By contrast, in *At*PIF6, the removal of the linker and the APB.B part more severely attenuated the affinity to the Pfr state to values between 200 and 700 nM. In addition, the affinity to the Pr state, non-detectable for the variants P6.100 and P6, increased as well. As a corollary, *At*PIF3 and *At*PIF6 variants lacking the APB.B segment exhibited closely similar $K_D$ values for a given construct topology. As a case in point, the P3.As and the P6.As variants, comprising 25 and 23 residues, respectively, both interacted with the *At*PhyB PCM with an affinity of ~700 nM in the Pfr and weaker than 10 µM in the Pr state. These data for P6As are consistent with a recent report that demonstrated light-dependent PPI for an *At*PIF6 construct of closely similar length and sequence[43]. Duplication of the APB.A segments in the *At*PIF3/6 backgrounds resulted in variants with affinities in the range of 200–400 nM for Pfr and weaker than 2 µM for Pr. We also analyzed several *At*PIF3/6 variants entirely lacking the APB.A segment or possessing shortened versions of it, neither of which showed any interaction with *At*PhyB PCM when probed by SEC nor by their effect on Pr reversion kinetics. In almost all these variants, fluorescence anisotropy failed to detect interactions either (Supplementary Fig. 6 and Table 1); merely, the P3.A19 and P6.A19 variants with C-terminally trimmed APB.A segments exhibited weak affinity for the Pfr state in the low micromolar range (Supplementary Fig. 6 and Table 1). In summary, these results

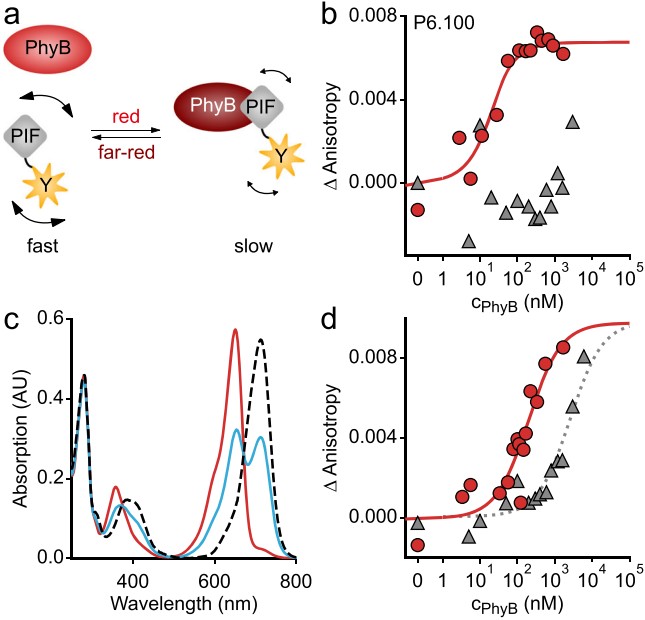

**Fig. 4** Quantitative analyses of the light-dependent protein:protein interaction between *At*PIF variants and the *At*PhyB PCM. **a** In its Pr state, the *At*PhyB PCM exhibits weak or no affinity to *At*PIF, but upon red-light exposure, the affinity is enhanced. Binding to the AtPhyB PCM increases the effective hydrodynamic radius of the *At*PIF variants and slows down rotational diffusion. In turn, the fluorescence anisotropy of an EYFP tag C-terminally appended to the *At*PIF increases. **b** Titration of 20 nM P6.100-EYFP with increasing concentrations of dark-adapted (gray) or red-light-exposed *At*PhyB PCM (red), as monitored by anisotropy of the EYFP fluorescence. Data points show mean of $n = 3$ biological replicates. The red line denotes a fit to a single-site-binding isotherm. **c** Absorption spectra of the *At*PhyB PCM in its dark-adapted Pr state (red line) and as a Pfr/Pr mixture following red-light exposure (blue). The dashed line denotes the absorption spectrum of the pure Pfr state, calculated according to ref. [42]. **d** As in **b** but for P3.100-EYFP rather than P6.100-EYFP. Experiments were repeated twice with similar results.

confirm the APB.A segment as the main interaction epitope in both *At*PIF3 and *At*PIF6. Intriguingly, *At*PIF6 differs from *At*PIF3 by higher affinity for Pfr and much reduced affinity for Pr. As the removal of the APB.B segment largely cancels these differences, we conclude that APB.B in *At*PIF6, but not in *At*PIF3, enhances the affinity for Pfr and diminishes that for Pr. In *At*PIF3, the N-terminal amino acids contribute to elevated basal affinity for Pr.

**Repurposing the *At*PhyB:*At*PIF interaction for optogenetics**. Through sequence variations and quantitative analyses, we generated modules for light-regulated PPIs spanning an affinity for the Pfr state from around 10 to 700 nM. We next investigated whether this set of novel *At*PIF variants can be leveraged for optogenetics in mammalian cells. In a first line of experiments, we embedded the variants into a previously reported system for red-/far-red-light-regulated gene expression that provides an in-cell readout of relative PPI affinities[36,44]. To this end, the *At*PhyB PCM was covalently attached to a VP16 *trans*-activating domain, and the different *At*PIF variants were linked to the E-protein DNA-binding domain, which binds to a cognate operator sequence upstream of a minimal promoter driving expression of secreted alkaline phosphatase (SEAP) (Fig. 5a). Through

light-induced *At*PhyB:*At*PIF interactions, the *trans*-activating domain localizes to the DNA-binding domain and the promoter and thereby induces SEAP expression. SEAP activity levels are quantified and normalized to the levels of constitutively expressed *Gaussia* luciferase to correct for variations of cell density, transfection efficiency, and overall expression. We found that the P3.100 and P6.100 reference constructs upregulated normalized SEAP expression by tenfold and fourfold, respectively, under red light compared to darkness when expressed in Chinese hamster ovary cells (CHO-K1). The comparatively small regulatory effect for P6.100 results from substantial basal SEAP expression. We then subjected all the *At*PIF3/6 variants we had previously characterized to the same analysis (Fig. 5b, c and Supplementary Fig. 7). Consistent with the above measurements, *At*PIF variants that lacked detectable interactions with the *At*PhyB PCM, e.g., P*x*.B and P*x*.A19, failed to stimulate reporter expression regardless of illumination. By contrast, variants that exhibited interactions with the *At*PhyB PCM were generally capable of inducing SEAP expression under red light, albeit to different degree. Overall, the expression levels observed for the individual *At*PIF variants scaled with binding affinity, in that low measured $K_D$ values correlated with strong SEAP activity. For instance, all *At*PIF3/6 variants containing the intact APB.A segment exhibited strong expression under red light. Whereas P6.100 suffered from relatively high basal expression, the shortened *At*PIF6 derivatives generally showed reduced SEAP expression in darkness, translating into much more pronounced regulatory effects. For instance, in the variant P6.A the SEAP expression increased by 43-fold under red light relative to darkness. Duplication of APB.A in the variants P6.AA and P3.AA elevated SEAP expression under red light and, to lesser extent, in darkness, thereby enhancing the regulatory effect. The overall higher SEAP expression under red light for these variants could reflect the binding of two *At*PhyB-VP16 modules to one P*x*.AA protein. However, we note that, under the conditions employed for the SEC analysis, we did not find evidence for simultaneous binding of two *At*PhyB PCM entities to the P*x*.AA variants. We also assessed the photoreversibility of the gene-expression systems based on the *At*PIF derivatives (Supplementary Fig. 8). When the cells were first exposed to red light for 24 h, followed by far-red illumination for another 24 h, they exhibited basal SEAP expression levels comparable to cells incubated in darkness throughout. Given that gene expression for the different sequence variations followed similar trends in both the *At*PIF3 and the *At*PIF6 backgrounds, we wondered whether the emerging underlying principles extend to other *At*PIF orthologs. We hence generated the corresponding sequence variations in the *At*PIF1 background and assessed their impact on light-regulated gene expression (Fig. 5d and Supplementary Fig. 7). Several of the resultant *At*PIF1 variants supported light-activated SEAP expression, although generally with slightly attenuated maximal levels and regulatory effects. Nonetheless, the *At*PIF1 variants conformed to the general activity pattern observed for the *At*PIF3/6 variants; specifically, only the *At*PIF1 variants preserving an intact APB.A segment were capable of upregulating SEAP expression under red light. Taken together, these experiments demonstrate the utility of the cellular set-up for the efficient appraisal of light-regulated PPIs in mammalian cells. By capitalizing on this set-up, we obtained derivative systems with enhanced dynamic range and reduced leakiness that outperformed the original reference systems.

In a second set of experiments, we deployed several of the newly generated *At*PIF6 variants for light-regulated recruitment of target proteins to the plasma membrane of NIH-3T3 cells. To this end, we equipped the *At*PhyB PCM with a C-terminal CAAX prenylation motif for membrane targeting and the *At*PIF6

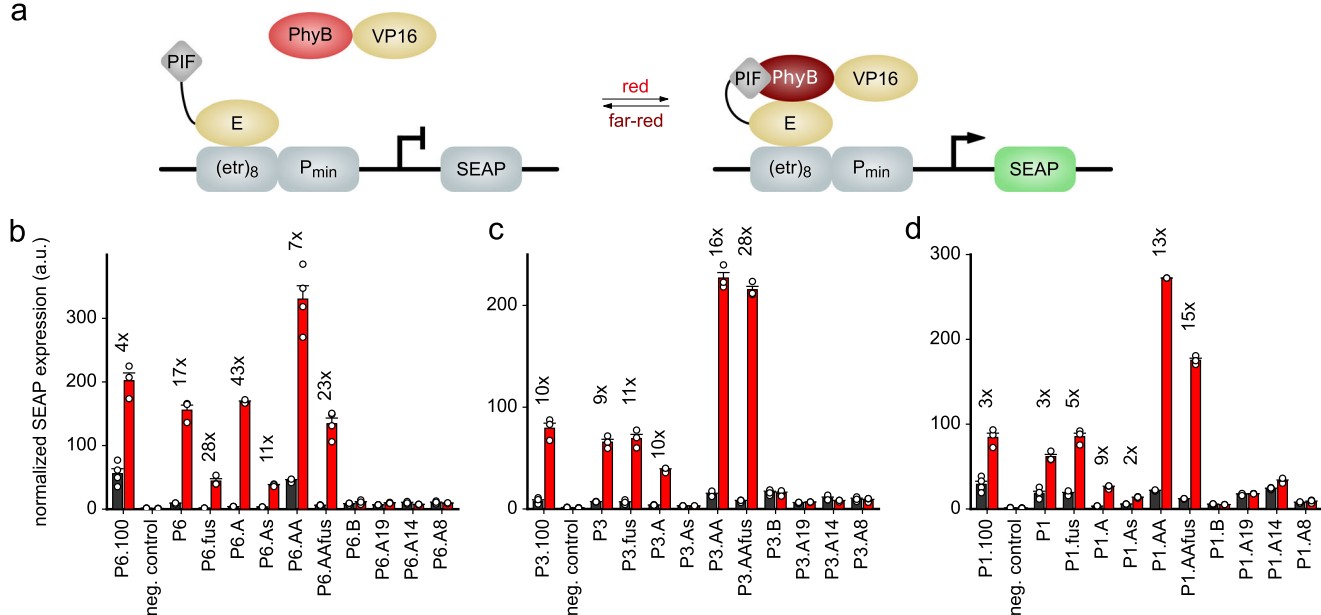

**Fig. 5** Harnessing the *At*PIF variants for the light-dependent regulation of gene expression in mammalian cells. **a** The *At*PhyB PCM and *At*PIF variants are connected to a VP16 *trans*-activating domain and an E-protein DNA-binding domain that binds to a synthetic promoter sequence. Red light promotes association of the *At*PhyB:*At*PIF pair and thereby activates the expression of a secreted alkaline phosphatase (SEAP) reporter gene. **b** SEAP expression was determined in Chinese hamster ovary cells (CHO-K1) for the diverse *At*PIF6 variants and normalized to the constitutive expression of *Gaussia* luciferase. Black and red bars denote mean ± SEM normalized SEAP expression for $n = 4$ independent biological replicates under dark or red-light conditions, respectively. Cells were kept in darkness for 24 h, supplemented with PCB, and then either kept in darkness for 24 h or illuminated for 24 h with 20 μmol m$^{-2}$ s$^{-1}$ 660-nm light. As a negative control, the reporter construct alone was transfected. The numbers above the bars indicate the factor difference between dark and red-light conditions for a given *At*PIF6 variant. **c** As **b** but for the *At*PIF3 variants. **d** As **b** but for the *At*PIF1 variants.

variants with an N-terminal EYFP tag[27,37,38] (Fig. 6a). Cell lines stably expressing both the *At*PhyB PCM and one of the *At*PIF6 variants, linked by an internal ribosome entry site (IRES), were created through lentiviral transduction. Cells were exposed to red (650 nm) and far-red light (750 nm), respectively, and the subcellular distribution of the EYFP-*At*PIF6 variants was monitored by fluorescence microscopy (Fig. 6b–e). Under far-red light, the reference variant P6.100 mostly localized to the cytoplasm, but under red light it partially translocated to the plasma membrane (Fig. 6c–f). Whereas the variants P6.A, P6.As, and P6.AA exhibited overall similar subcellular distribution under red and far-red light as P6.100, the variant P6.fus failed to show any light response of subcellular localization. Although subtle performance differences between the individual *At*PIF6 variants cannot be ruled out, these are exceeded by the cell-to-cell variability of light-dependent translocation (Fig. 6f). Nonetheless, the experiments show that the new *At*PIF6 variants with a much smaller footprint support light-regulated plasma membrane recruitment at similar efficiencies as the reference P6.100. This notion is further supported by the overall comparable expression level of the *At*PIF6 variants and its effect on the magnitude of light-regulated membrane recruitment (Fig. 6g).

## Discussion
In this study, we have dissected the light-regulated PPIs between the *At*PhyB PCM and the *At*PIFs 3 and 6, which underpin diverse adaptive responses in planta and multiple applications in optogenetics. To this end, we implemented a set of complementary experimental approaches ranging from SEC and fluorescence anisotropy to reporter assays in mammalian cells that deliver

both qualitative and quantitative information on the PPIs. At a qualitative level, these assays consistently showed the APB.A segment to be necessary and sufficient for *At*PhyB-PCM interactions, in line with previous reports[24]. By contrast, the APB.B segment alone did not promote detectable interactions. Our quantitative analyses put concrete numbers on the affinity of the *At*PhyB:*At*PIF3/6 pairs, information that hitherto was largely lacking. Strikingly, P6.100 exhibited a $K_D$ of only ~10 nM for *At*PhyB PCM in its Pfr state but entirely lacked interaction with the Pr state, from which we estimate an at least 1000-fold affinity difference. By contrast, the light dependence of the P3.100: *At*PhyB-PCM interaction was less pronounced, with dissociation constants of ~200 nM in the Pfr state and low micromolar in the Pr state. We tied the more stringent red-light response in *At*PIF6 to its APB.B segment, which enhances affinity for the Pfr state of the *At*PhyB PCM while simultaneously attenuating basal affinity for the Pr state. We speculate that these inherent differences between AtPIF3 and *At*PIF6 might reflect their natural roles in planta. Whereas *At*PIF3 predominantly serves as a negative regulator of photomorphogenesis by modulating the abundance of *At*PhyB[45–47], *At*PIF6 acts as a positive regulator by inhibiting hypocotyl elongation under red light, at least when over-expressed[48]. To prevent untimely inhibition of hypocotyl growth, a more stringent light response with very low basal affinity in Pr may be required for this particular PIF. Recently, it has been reported that PIFs, and in particular *At*PIF3, are constantly turned over both in darkness and under red light as a mechanism of achieving optimal levels for tight regulation of the skotomorphogenic and photomorphogenic responses[14]. A more permissive binding of *At*PIF3 to the Pr state of *At*PhyB as observed here

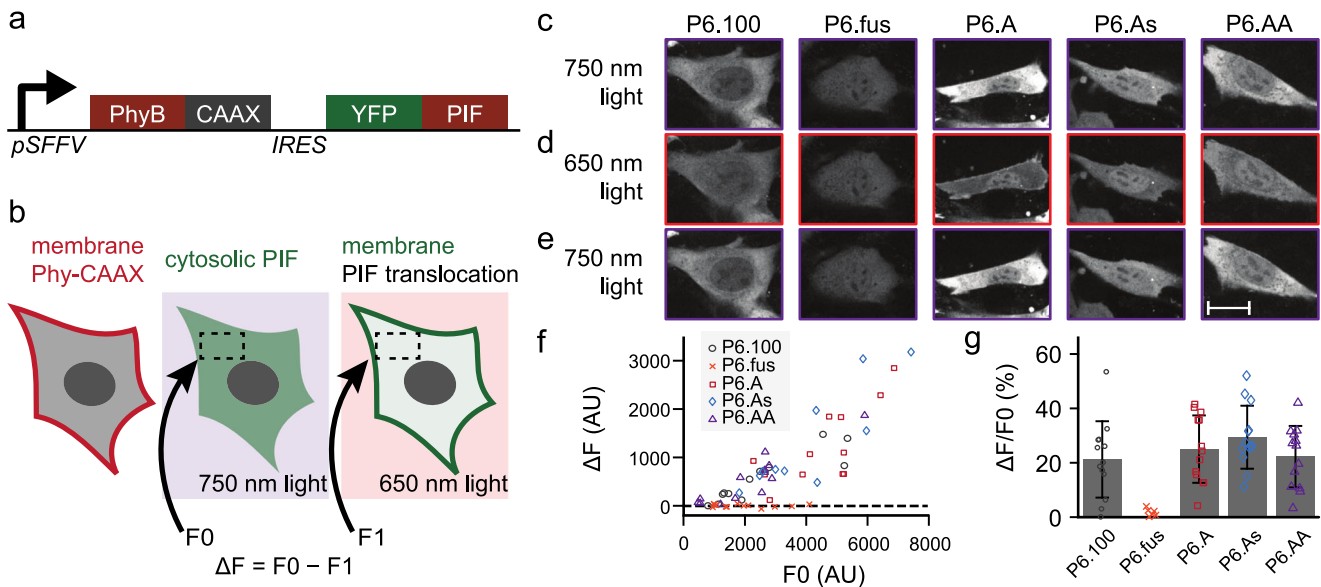

**Fig. 6 Photoreversible recruitment to the plasma membrane. a** *At*PhyB and one of the several new *At*PIF6 variants, equipped with a C-terminal CAAX prenylation motif or an N-terminal EYFP tag, respectively, were encoded on a bicistronic vector with an intervening IRES sequence and expressed in NIH-3T3 cells. **b** Owing to the CAAX tag, *At*PhyB localizes to the plasma membrane, while the EYFP-*At*PIF6 variants shuttle between cytosol and plasma membrane as a function of light. Under far-red light (750 nm), the EYFP-*At*PIF6 variants exhibit cytosolic localization; under red light (650 nm), they can bind to *At*PhyB and translocate to the membrane. **c–e** Fluorescence micrographs of NIH-3T3 cells expressing *At*PhyB-CAAX and different EYFP-*At*PIF6 variants under far-red light (**c**), after red-light exposure (**d**), and after additional exposure to far-red light (**e**). The scale bar denotes 20 μm. **f** The relative depletion of cytosolic EYFP fluorescence under red light compared to far-red light for the EYFP-*At*PIF6 variants. Data represent mean ± SD of $n \geq 12$ individual cells. **g** Dependence of the relative fluorescence change on the overall EYFP-*At*PIF6 expression level.

might facilitate the regulation of PIF abundance in darkness. This concurs with reports that *At*PhyB mediates phosphorylation by PPK-family kinases and subsequent degradation of *At*PIF3 in both the Pr and Pfr states[49]. The differential affinities of the individual PIFs might therefore contribute to the fine-tuning of physiological responses[14,49]. In fact, our study now provides a means of gradually adjusting the interaction strength of a given PIF, which could benefit the analysis of signal transduction mechanisms in planta. In a similar vein, the quantitative data on the *At*PhyB:*At*PIF PPI may help rationalize the phenotypes of pertinent *pif* mutant alleles. Finally, the comparatively smaller regulatory effect in *At*PhyB:*At*PIF3, compared to *At*PhyB:*At*PIF6, may explain why this PPI pair proved inferior for generic optogenetic applications[27].

By deconstructing and quantitatively analyzing *At*PIF3/6, we devised a suite of interaction modules with several beneficial traits (Table 1): First, the *At*PIF variants span an affinity range from 10 to 700 nM, thus enabling the precise tuning of light-regulated PPIs as demanded by a specific application. Second, the *At*PIFs can be reduced to around 23–25 residues while largely retaining light-regulated PPIs with the *At*PhyB PCM. As we demonstrate, the smaller size facilitates the construction of tandem repeats of the APB.A motif, which, depending upon context, may enhance light-dependent responses. Third, the reduction in size also affected the oligomeric state of the *At*PIFs, which are homodimeric at full length[41] but predominantly monomeric in several of the truncated variants studied presently. As we showcase for the scenarios of light-regulated gene expression and membrane recruitment, the set of novel *At*PIF variants can indeed improve absolute activity and degree of light regulation in optogenetics. As a case in point, despite stringently light-regulated PPIs with the *At*PhyB PCM, the original P6.100 variant promoted substantial basal gene expression in darkness, thus degrading the regulatory effect of light. We tentatively ascribe the relatively poor performance of P6.100 to its

high Pfr-state affinity; even limited population of the *At*PhyB Pfr state, e.g., due to light pollution or temperature changes[50], may hence activate the PPI to considerable extent and over prolonged periods[39]. In support of this notion, the attenuation of the Pfr-state affinity in the shortened *At*PIF6 variants led to reduced basal activity and enhanced regulatory efficiency. Duplication of the APB.A segment improved the performance for light-regulated expression, although the Pfr-state affinity of the P*x*.AA variants is almost unchanged relative to the corresponding P*x*.A variants. We hence ascribe this improvement to avidity and cooperativity effects. Our analyses readily extended to the *At*PIF1 context, where shortened variants exhibited similar patterns of activity and light regulation as the *At*PIF3/6 variants (cf. Fig. 5b–d). We speculate that the underlying principles can be generalized to APB-containing PIF proteins from *A. thaliana* and other plants[51,52]. The performance of individual *At*PIF variants in a given experiment can considerably vary and may be difficult to gauge upfront, not least because it likely depends on application context. We thus consider it an advantage to have now a set of *At*PIF variants with known interaction strengths and varying properties. With this suite of *At*PIF variants in hand, additional processes may be unlocked for optogenetic control by red and far-red light. As recently summarized[53], numerous cellular parameters and pathways depend on PPIs and can thus be controlled by certain photoreceptors that associate or dissociate under blue light. The underlying regulatory strategy should readily extend to the present *At*PhyB:*At*PIF pairs and thereby to red and far-red light. Other potential use cases for the new *At*PIF variants include immunoreceptor signaling[30] and light-regulated biomaterials[54]. As one shortcoming, optogenetic applications of plant Phys currently require the exogenous addition of PCB or PΦB chromophores, which do not widely occur outside cyanobacteria and plants. This contrasts with bacterial Phys, which utilize biliverdin (BV) that is available in mammals as a heme degradation

product[55–58]. In particular, a recently described bacterial Phy undergoes PPIs depending on red and far-red light and has been harnessed for light-regulated gene expression[59–61]. The reliance on BV in this system obviates exogenous chromophore addition, which may prove advantageous for applications in vivo.

In summary, we have constructed and characterized a toolkit of novel AtPIF variants with varying interaction strength, size, and oligomeric state. Beyond application in optogenetics, the availability of these variants also stands to benefit the biophysical analyses of the Phy:PIF interaction. Although previous studies had localized this interaction to the N-terminal extension of Phys, atomically resolved information on the Phy:PIF complex is lacking[40,62–64]. Minimized AtPIFs may well facilitate X-ray crystallographic analysis and thus pave the way toward elucidation of the complex structure. Moreover, the qualitative and quantitative interaction assays presently established can be deployed to chart Phys and interacting factors from A. thaliana and other plants.

## Methods

**Molecular biology and protein purification.** Genes encoding A. thaliana PhyB PCM (residues 1–651), PIF3 (1–100), and PIF6 (1–100) were synthesized with codon usage adapted for expression in E. coli (GeneArt, Invitrogen, Regensburg, Germany). Via Gibson assembly[65], the AtPhyB PCM was furnished with a C-terminal hexahistidine tag and subcloned onto the pCDFDuet1 vector (Novagen, Merck, Darmstadt, Germany) under control of a T7-lacO promoter; the plasmid, designated pDG282, additionally harbors a bicistronic cassette of Synechocystis sp. heme oxygenase 1 and pcyA[66], also under the control of T7-lacO. For the expression of AtPIF3/6, the corresponding genes were subcloned onto a pET-19b vector (Novagen) under the control of a T7-lacO promoter by Gibson assembly or AQUA cloning[67] and thereby equipped with an N-terminal His$_6$-SUMO tag[68] and a C-terminal EYFP tag, attached via a short linker (DSAGSAGSAG). For interaction studies in bacterial lysate, the AtPIF3/6 genes were subcloned onto a pET-28c vector (Novagen) under the control of a T7-lacO promoter, again with C-terminal linkers and EYFP. Variants of the AtPIF proteins were generated in both plasmid contexts, and the identity of all constructs was confirmed by Sanger DNA sequencing (GATC, Konstanz, Germany or Microsynth Seqlab, Göttingen, Germany).

For AtPhyB expression, the plasmid pDG282 was transformed into the E. coli BL21(DE3) strain. Transformant cells were grown in 2× 1000 mL terrific-broth (TB) medium, supplemented with 100 μg mL$^{-1}$ streptomycin, at 37 °C in darkness until an optical density at 600 nm (OD$_{600}$) of 0.6–0.8 was reached. δ-Aminolevulinic acid was added at 0.5 mM to assist chromophore production[69], and the expression was induced by adding 1 mM β-D-1-thiogalactopyranoside (IPTG). Cultivation continued overnight at 18 °C, before cells were harvested by centrifugation, resuspended in lysis buffer [50 mM Tris/HCl pH 8.0, 20 mM NaCl, 20 mM imidazole; supplemented with protease inhibitor mix (cOmplete Ultra, Roche Diagnostics, Mannheim, Germany)], and lysed by sonification. The cleared lysate was purified by immobilized ion affinity chromatography (IMAC) on Protino Ni-NTA 1 mL columns (Macherey-Nagel, Düren, Germany) and eluted with a linear imidazole gradient from 20 to 500 mM. Elution fractions were analyzed by denaturing polyacrylamide gel electrophoresis (PAGE), where 1 mM Zn$^{2+}$ was added to enable detection of covalently incorporated bilin chromophores via zinc-induced fluorescence[70]. Suitable fractions were pooled and dialyzed overnight into AEX buffer (20 mM Tris/HCl pH 8.0, 50 mM NaCl, 5 mM 2-mercaptoethanol), applied to a HiTrap Q HP 1 mL anion-exchange column (GE Healthcare Europe GmbH, Freiburg, Germany), and eluted using two successive linear gradients from 50 to 300 mM NaCl and from 300 to 500 mM. Eluted fractions were analyzed by PAGE, appropriately pooled, dialyzed against storage buffer [10 mM Tris/HCl pH8, 10 mM NaCl, 10 % (w/v) glycerol], and stored at −80 °C.

Purification of the AtPIF3/6-EYFP variants employed a similar protocol with the following differences. No δ-aminolevulinic acid was added, and incubation after induction continued at 16 °C for 40 h. Following IMAC, the N-terminal His$_6$-SUMO was cleaved overnight at 4 °C during dialysis into 50 mM Tris/HCl pH 8.0 and 20 mM NaCl using SENP2-protease. The His$_6$-SUMO tag was removed by IMAC, and the flow-through containing the AtPIF3/6 construct was collected and analyzed by PAGE. Depending upon purity, the proteins were optionally further purified by anion-exchange chromatography as described above. Pure AtPIF3/6-EYFP variants were dialyzed into storage buffer and stored at −80 °C. An analysis by denaturing PAGE of the purified AtPIF3/6-EYFP constructs and the AtPhyB PCM is shown as Supplementary Fig. 9.

**Spectroscopic analysis.** The concentration of purified AtPhyB PCM and the AtPIF3/6-EYFP variants were determined at 22 °C by absorption measurements on an Agilent 8453 UV-visible spectrophotometer (Agilent Technologies, Waldbronn, Germany). In case of the AtPIF3/6-EYFP variants, a molar extinction coefficient at 513 nm of 84,300 M$^{-1}$ cm$^{-1}$ was used[71]. Photoreversible

Pr ↔ Pfr conversion of AtPhyB PCM was ascertained by illumination with light-emitting diodes (LED) with emission wavelengths of 650 ± 15 nm (5.6 μW cm$^{-2}$) and 720 ± 15 nm (0.7 μW cm$^{-2}$), respectively. Spectra recorded after illumination revealed isosbestic points at 374 and 672 nm. Absorption spectra were also recorded after denaturation in 6.5 M guanidinium hydrochloride. By referencing to the previously reported extinction coefficient for PCB under these conditions[72], we calculated an extinction coefficient at the isosbestic point 672 nm for AtPhyB PCM in its native state of 47,600 M$^{-1}$ cm$^{-1}$. The fraction of AtPhyB PCM in the Pfr state upon saturating red-light illumination (640 nm) was determined as described in ref. [42].

**Interaction assay in bacterial lysate.** pET-28c plasmids harboring AtPIF3-EYFP or AtPIF6-EYFP variants were transformed into chemically competent BL21(DE3) cells. Three replicate clones were used to inoculate 3× 5 mL TB medium supplemented with 50 μg mL$^{-1}$ kanamycin. Cultures were incubated at 37 °C up to an OD$_{600}$ of 0.6–0.8, at which point temperature was lowered to 16 °C and expression was induced by addition of 1 mM IPTG. Incubation continued overnight, and cells were harvested by centrifugation at 3000 × g for 10 min. Pelleted cells were resuspended in 300 μL lysis buffer [1× FastBreak Cell Lysis Reagent (Promega GmbH, Mannheim, Germany), 10 μg mL$^{-1}$ DNaseI (PanReac AppliChem, Darmstadt, Germany), 200 μg mL$^{-1}$ lysozyme (Sigma-Aldrich, Darmstadt, Germany)] and rotated at 22 °C for 10 min. Cell debris was removed by centrifugation at 186,000 × g for 45 min using an Optima MAX-XP Ultracentrifuge (Beckman-Coulter, Krefeld, Germany). The concentration of a given AtPIF3/6-EYFP variant in the lysate was determined by absorption measurements at 513 nm using a CLARIOstar microtiter plate reader (MTP) (BMG Labtech, Ortenberg, Germany). AtPhyB PCM at 2.5 μM concentration was mixed with a threefold molar excess of the AtPIF3/6-EYFP variants in 384-well clear MTPs (Thermo Fisher Scientific, Waltham, USA). After illumination with red light (650 ± 15 nm, 5.6 μW cm$^{-2}$) for 4 min, the MTPs were covered with a clear lid, and absorption at 720 and 850 nm was measured every 5 min at 28 °C in an Infinite M200 PRO plate reader (Tecan, Männedorf, Switzerland) for 12 h. After background correction, data at 720 nm were normalized to the signal of the L-EYFP (Supplementary Table 1) negative control, and the relative initial velocity was determined over the data acquired during the first 4 h.

**Interaction assays with purified components.** Size-exclusion chromatography: The light-dependent interaction between AtPhyB PCM and the AtPIF3/6-EYFP variants was assessed by gel filtration chromatography using a Superdex 200 Increase 10/300 GL (GE Healthcare) column on an ÄKTApure system, equipped with multi-wavelength detection (GE Healthcare). To this end, a mixture of 50 μM AtPhyB-PCM and 10 μM PIF-EYFP in 67 mM sodium phosphate buffer pH 8.0 and 200 mM NaCl was prepared and illuminated with 650- or 720-nm light for 2 min before sample application. Twenty-five microliters of this mixture was applied to the column and separated at a constant 0.75 mL min$^{-1}$ flow rate. Absorption of EYFP and the AtPhyB PCM was measured at 513 and 650 nm, respectively. All proteins were also tested individually, where the AtPIF3/6-EYFP and EYFP samples were not illuminated prior to application.

Fluorescence anisotropy: AtPhyB PCM was illuminated with 640- or 750-nm light for 2 min immediately prior to the experiment (640 ± 15 nm; 65 μW cm$^{-2}$ and 750 ± 15 nm; 420 μW cm$^{-2}$). Samples containing 20 nM AtPIF3/6-EYFP and increasing AtPhyB-PCM concentrations between 0 and 10 μM were prepared in 20 mM HEPES/HCl pH 7.3, 10 mM NaCl, and 100 μg mL$^{-1}$ bovine serum albumin, transferred into black 384-well MTPs (Brand, Wertheim, Germany), and illuminated with 640- or 750-nm light. Fluorescence anisotropy of EYFP fluorophore was measured on a CLARIOstar MTP reader (BMG Labtech) with an excitation wavelength of 482 ± 16 nm, a 504-nm long-pass dichroic filter, and a detection wavelength of 530 ± 40 nm. The fluorescence gains for the horizontal and vertical detection channels were adjusted to a fluorescence anisotropy value of 0.315, as determined for EYFP with an Olis DSM 172 spectrophotometer (On-Line Instrument Systems, Bogart, USA). Anisotropy data were evaluated with the Fit-o-mat software[73] using a single-site binding isotherm:

$$r = r_0 + r_1 \frac{[\text{PhyB}]}{[\text{PhyB}] + K_D}$$

where $r$ represents the anisotropy of the PIF-EYFP fluorescence, [PhyB] is the concentration of the AtPhyB PCM in either the Pr or Pfr state, and $K_D$ is the dissociation constant. For the case of strong binding exhibited by the variants P6.100 and P6, we used a modified single-site binding isotherm that takes into account that the relevant [PhyB] concentrations are on the same order of magnitude as the constant concentration $c_{total}$ of the PIF-EYFP protein:

$$r = r_0 + r_1/2 \times \left\{ 1 + [\text{PhyB}]/c_{total} + K_D/c_{total} \right.$$
$$\left. - \sqrt{(1 + [\text{PhyB}]/c_{total} + K_D/c_{total})^2 - 4[\text{PhyB}]/c_{total}} \right\}$$

**Light-regulated gene expression in mammalian cells.** The split transcription factor system for light-controlled gene expression in eukaryotic cells was based on a previously reported set-up[36,44]. To allow ratiometric analysis, this earlier set-up

was expanded by cloning the *Gaussia* luciferase under the control of a constitutive promoter onto the same plasmid as the SEAP reporter gene. For testing of the *At*PIF variants, *At*PIF6 (1–100) was replaced by the corresponding *At*PIF1/3/6 derivatives. CHO-K1 (DSMZ, Braunschweig, Germany) were cultivated in HAM's F12 medium (PAN Biotech, Aidenbach, Germany; no. P04–14500) supplemented with 10% (v/v) tetracycline-free fetal bovine serum (PAN Biotech; no. P30–3602; batch no. P080317TC) and 1.4% (v/v) streptomycin (PAN Biotech; no. P06–07100). In all, $5 \times 10^4$ CHO-K1 cells were transfected using polyethyleneimine (PEI; Polysciences Inc. Europe, Hirschberg, Germany; no. 23966–1)[74]. DNA (0.75 µg) was diluted in 50 µL OptiMEM (Invitrogen, Thermo Fisher Scientific) and mixed with a PEI/OptiMEM mix (2.5 µL PEI solution in 50 µL OptiMEM). The DNA–PEI mix was added to the cells after 15 min of incubation at room temperature. At 4 h post-transfection, the medium was exchanged. CHO-K1 cells were transfected with the reporter plasmid etr8-CMVmin-SEAP-BGH-SV40-Gaussia (pPF035) and the different *At*PhyB:*At*PIF variants. All plasmids were transfected in equal amounts (w/w). At 24 h post-transfection, the cells were supplemented with 15 µM phycocyanobilin (24 mM stock solution in DMSO; Frontier Scientific, Logan, UT, USA; no. P14137) and incubated for 1 h. The cells were then illuminated with 660-nm light for 24 h at an intensity of 20 µmol m$^{-2}$ s$^{-1}$, delivered by custom-built LED light boxes[36], or kept in darkness. As a negative control, the reporter construct alone was transfected. Photoreversibility was tested by incubating cells for 24 h under red light, followed by exchange of the media and incubation under far-red light for 24 h. In parallel, cells were incubated in darkness for 48 h with media exchange after 24 h. Exchange of media and other cell handling was done under 522-nm safe light, to prevent inadvertent actuation of the light-sensitive systems.

*SEAP activity assay*: The supernatant of transfected cells was transferred to 96-well round-bottom MTPs and incubated at 68 °C for 1 h to inactivate endogenous phosphatases. Afterwards, 80 µL of the supernatant were transferred to 96-well flat-bottom MTPs, and per well 100 µL SEAP buffer [20 mM homoarginine, 1 mM MgCl$_2$, 21% (v/v) diethanolamine] was added[36]. After addition of 20 µL 120 mM para-nitrophenyl phosphate, the absorption at 405 nm was measured for 1 h using a BMG Labtech CLARIOstar or a TriStar2 S LB 942 multimode plate reader (Berthold Technologies, Bad Wildbad, Germany)[36]. Outliers were statistically determined and excluded[75].

*Gaussia luciferase assay*: Twenty microliters of the supernatant of the transfected cells were transferred to a 96-well white MTP and diluted in 60 µL phosphate-buffered saline (PBS; 2.68 mM KCl, 1.47 mM KH$_2$PO$_4$, 8.03 mM Na$_2$PO$_4$, 137 mM NaCl). After addition of 20 µL coelenterazine (472 mM stock solution in methanol, diluted 1:1500 in PBS; Carl Roth, Karlsruhe, Germany, no. 4094.4), the luminescence was measured for 20 min using TriStar2 LB 941 or LB 942 multimode plate readers.

**Light-mediated membrane recruitment in mammalian cells**. For each *At*PIF6 variant tested, a lentiviral vector (pHR) was constructed containing a membrane-bound *At*PhyB PCM (PHY-CAAX, residues 1–650) and a YFP-conjugated *At*PIF6 variant. An IRES was introduced between the two coding sequences to ensure regulation of dual expression. Lentivirus was created by transfecting HEK-293T cells with pHR constructs and harvesting filtered media 48 h post-transfection. Mouse fibroblasts (NIH-3T3) were cultured in Dulbecco's Modified Eagle Medium (DMEM) containing 10% (v/v) fetal bovine serum. Fibroblasts were treated with lentivirus containing the constructs of interest. For all fibroblast experiments, cells were cultured in a 96-well glass-bottomed plate. Wells were pretreated with fibronectin for 30 min, following which fibronectin was aspirated and cells were plated and spun down for 5 min at 800 rpm. Cells were plated in 96-well glass-bottom plates and allowed to adhere for at least 12 h. Imaging was performed using a ×60 oil immersion objective (NA 1.4) on a Nikon TI Eclipse microscope with a CSU-X1 confocal spinning disk, an EM-CCD camera, and appropriate laser lines, dichroics, and filters. DMEM was supplemented with phycocyanobilin 30 min prior to the start of the experiment. Cells were exposed to infrared light followed by red light to cause membrane recruitment and the resulting change in cytoplasmic fluorescence was measured using ImageJ by selecting a cytoplasmic region and computing the average pixel intensity before and after photostimulation. The change in cytoplasmic YFP-PIF level was normalized to the total YFP-PIF fluorescence in the nucleus under infrared conditions, to normalize to total expression level differences caused by lentivirus. In these experiments, light was delivered through the microscope using a Mightex Polygon digital micromirror device (DMD), X-Cite XLED1 LED light sources at 635 ± 20 and 730 ± 20 nm, and a ×40 objective lens. The duration of LED illumination was 1 min. To estimate the light dose delivered to the cell, we measured the light intensity using a ThorLabs power meter (PM100D) when the DMD was set to 100% transmission and obtained 100 µW for 635-nm light and 20 µW for 730-nm light, over a field of view of about 100 µm squared. For all experiments, we set the DMDs to 5% dithering (so each region was only illuminated for 5% of the time), translating into a final calculated intensity of 5 µW 635-nm light and 1 µW of 730-nm light. The light was delivered over an approximately 100 µm × 100 µm field of view, leading to an overall LED power density of 50 mW cm$^{-2}$ at 635 nm and 10 mW cm$^{-2}$ at 730 nm. Notably, these values are slightly higher but of comparable magnitude to those used by Pathak et al. for the *At*PhyB:*At*PIF3/6 system in the context of light-regulated gene expression[76].

**Statistics and reproducibility**. Data are reported as mean ± SD or as mean ± SEM of $n \geq 3$ biologically independent replicates. Details are specified in the legends to the figures and tables. All experiments could be reproduced with similar results.

**Reporting summary**. Further information on research design is available in the Nature Research Reporting Summary linked to this article.

## Data availability
The data underlying Figs. 2–6 are available in Supplementary Data 1. All data that support the findings of this study are available from the corresponding author upon reasonable request.

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

## Acknowledgements

We thank members of our laboratories for support and comments; Dr. J. Casal for fruitful discussion; Dr. B. Höcker for chromatography use; Dr. T. Scheibel for ultracentrifuge use; and Dr. M. Hörner and Dr. W. Weber for generously providing PhyB expression plasmids. Funding through the Deutsche Forschungsgemeinschaft (grants MO2192/7-1 to A.M., ZU259/2-1 to M.D.Z., and ES152/16 to L.-O.E.), under Germany's Excellence Strategy CEPLAS EXC2048/1 (to M.D.Z., ID 390686111), by NIH grant DP2EB024247 (to J.E.T.), by NIH training grant T32GM007388 (to S.G.J.), and by the European Commission – Research Executive Agency (H2020 Future and Emerging Technologies (FET-Open) Project ID 801041 CyGenTiG to M.D.Z.) is appreciated. J.R.W.K. gratefully acknowledges support through the ENB program "Biological Physics."

## Author contributions

D.G. designed and cloned the *At*PIF variants, expressed them and *At*PhyB, analyzed light-dependent interactions in lysate and for purified proteins, cloned the constructs for light-regulated gene expression, and analyzed data. P.F. performed and evaluated experiments on light-regulated gene expression. S.G.J. conducted and evaluated experiments on light-regulated membrane recruitment. J.R.W.K. cloned and expressed several *At*PIF variants and analyzed their interactions with *At*PhyB in lysate. L.-O.E. advised on experimental design. J.E.T. supervised experiments on light-regulated membrane recruitment. M.D.Z. conceived the project and designed and supervised research. A.M. conceived the project and designed and supervised research. D.G., P.F., M.D.Z., and A.M. wrote the manuscript with input from all authors.

## Competing interests

The authors declare no competing interests.
