## [Peer Review File · Communications Biology]

Reviewers' comments:

Reviewer #1 (Remarks to the Author):

A well-written manuscript and well-executed study focusing on the detailed characterisation and utilisation of light-driven protein-protein interactions between the plant red/far-red light receptor, phytochrome phyB, and the phytochrome interacting transcription factors (PIFs).

Golonka and co-authors dissect the site of interaction between a series of PIF and phyB fragments and uncover not only novel PIF-specific and phyB peptides that are necessary and sufficient for association as well as red light induced responses. The authors focus on the differentially regulated interaction between PIF3 and PIF6 with phyB and build novel proof-of-concept synthetic constructs that clearly demonstrate light-regulated gene expression and membrane targeting in orthogonal systems using mammalian cells.

Data presentation, analysis and interpretation are solid and the conclusions fully justified.

This study will undoubtedly open new avenues in addressing fundamental plant biology questions.

Furthermore, the use of such novel, tuneable optogenetic tools will have wide applications in research and medical fields.

Minor points:

1. If possible, could the authors include as supplemental figures images of SDS-PAGE gels showing the molecular weight, integrity and relative abundance of the E.coli expressed segments examined (and western blots of the mammalian expressed proteins, if a tag is present)?
2. The manuscript could benefit from further highlighting/expanding the discussion on:
 - a) how these findings could be potentially utilised to enhance our understanding of the function of the native plant proteins (PIF-phyB)
 - b) specific examples of novel, optogenetic applications of the toolkit presented in this manuscript in addition to the ones stated.

Reviewer #2 (Remarks to the Author):

The red and far-red light regulated binding between *Arabidopsis thaliana* phytochrome B (AtPhyB) and *A. thaliana* phytochrome-interacting factors (AtPIFs) has been applied in optogenetics to study the dynamic control of cellular signaling, and this approach offers fast, light-switchable and reversible regulations with high spatiotemporal resolution. The submitted manuscript by Golonka and co-workers established a screen approach and quantitatively analyzed the interaction between the N-terminal part of AtPhyB, AtPhyB PCM, and different variants of the N-terminal APB motif within AtPIF3 and AtPIF6, and they narrowed down the minimal AtPIF3/6 fragments responsible for interaction with AtPhyB PCM to 25 and 23 residues of the APB.A region respectively. They further deployed these AtPIF3/6 variants for usage in optogenetic control of gene expression and membrane recruitment in CHO-K1 and NIH-3T3 cells respectively. Overall the results reported in the manuscript provided valuable quantitative data about the AtPhyB:AtPIF interaction and the identified shortened AtPIF3/6 variants can have potential usage in optogenetics to optimize the light-gated gene expression and protein-protein interactions. The work is a good contribution to the field and will be interesting to a broad readership. I am overall enthusiastic about its publication, but do have the following minor comments.

Point 1.

The light-regulated AtPhyB:AtPIF interaction offers a reversible regulation, the system can be switched ON and OFF. In Fig. 5, the authors tested the interaction between the AtPIF variants and AtPhyB PCM

in a previously reported system (Müller et al 2014 Nature Protocols and Müller et al 2014 Molecular BioSystems), and the system has been shown to be reversible in previous reports. However results presented in Fig 5 did not provide data of the OFF switch. The dark and red-light conditions as described in line 508-509 were cells kept in two conditions respectively. How will the SEAP gene expression respond to a far-red light illumination after the red light illumination? Can the truncated variants of AtPIFs influence the far-red light caused OFF switch in vivo? Can the extremely higher SEAP expression caused by the Px.AA variants be reversed by far-red light? And also, how long does the red light illumination and darkness last? There is no "indicated period of time" (line 508) in Fig. 5 and supplementary Fig. 7. What is neg. control in Fig. 5?

Point 2.

Fig. 6 used a similar system as Levskaya et al 2009 Nature. The 2009 report indicated the tandem PAS repeats of PhyB C-terminal OPM region is required for the reversible membrane recruitment of PIF6 in vivo. Since the AtPhyB PCM used in Fig 6 does not contain the tandem PAS repeats and the results shown went through a far-red light illumination followed by a red light illumination, I am wondering can the membrane recruitment of AtPIF6 variants be reversed by another far-red light illumination after the red light. And also what is the intensity and duration of the red and far-red light used in Fig. 6 results?

Point 3.

Multiple evidence in the current manuscript seems to show the linker region between APB.A and APB.B or part of the linker region are important for the AtPhyB PCM and AtPIF3/6 or AtPIF1 variants interaction: supplementary Fig. 4 D, F, H and I; the Pfr state KD values for P3.As and P6.As (compared to the Pfr state KD values for P3.A and P6.A respectively) in Table 1 and the Pfr state KD values for P6. fus (compared to the Pfr state KD values for P6) in Table 1; P6.As compared to P6.A in Fig. 5B, P3.As compared P3.A in Fig. 5C and P1.As compared to P1.A in Fig. 5D. Do the authors have any explanation about this?

Point 4.

The names of P3.P1L, P6.P1L, P3.SL and P6.SL in supplementary Table 1 and supplementary Fig.7 are not consistent with the main text and Fig. 1E.

Reviewer #3 (Remarks to the Author):

The manuscript submitted by Golonka et al. describes basically about the mapping of APB (active phytochrome B (phyB) binding) motifs of PIFs (phytochrome-interacting factors) and its applications to optogenetics. Specifically, the authors generated tens of APB variants using three PIFs (PIF1, PIF3, PIF6) and investigated their interactions with the N-terminal photosensory core module (PCM) of *Arabidopsis thaliana* phyB (AtphyB). For this, they developed and used a screening platform to analyze protein-protein interactions (PPIs) between AtphyB-PCM and the APB variants of PIFs, in which thermal reversion (previously known as dark reversion) of purified AtphyB-PCM was monitored in the presence of the APB variants. Moreover, the authors analyzed oligomeric states of the APB variants by size-exclusion chromatography (SEC) and also analyzed the PPIs quantitatively by determining dissociation constants (Kd values) using fluorescence anisotropy measurements. At the end, they applied the APB variants for the tools of optogenetics such as light-regulated gene expression and photoreversible recruitment to plasma membrane.

Overall, the novelty of this study is not strong, because optogenetics studies such as light-regulated gene expression and photoreversible recruitment to plasma membrane are already reported (see the references in the manuscript) and the APB sequence of PIF6 consisting of 22 amino acids is already known to be sufficient for the interaction with AtphyB (see reference 38). However, as the authors mentioned, systemic studies of PPIs between AtphyB and PIFs are not reported, so the results of this manuscript might be useful for the future study of phytochrome-PIFs interactions and also

optogenetics study especially with plant phytochromes.

The contents and organization of the manuscript are easy to follow the work done for this study. However, here are some comments to improve the manuscript.

1. There are no data to see the quality of proteins used in this study. Thus, it would be better to include SDS-PAGE gels to show purified proteins used in this study.

In addition, as the authors mentioned that the C-terminal output module (OPM) contributes to light-regulated PPIs and is possibly important for eliciting physiological responses, is there any result of PPIs between full-length or OPM of AtPhyB and the APB variants?

In Suppl. Fig. 7, P3.SL and P1.SL showed light-dependent expression but P6.SL did not. Is there any explanation for this result?

2. It would be recommended to use "phyB", instead of "PhyB", for phytochrome B, according to the nomenclature of phytochromes (Plant Cell, 1994, 6:468-471). Similarly, use "phy" instead of "Phy" for phytochrome, and AtphyB and AtphyA instead of AtPhyB and AtPhyA etc.

3. It might be better to mention about the serine/threonine kinase activity observed in plant phytochromes in the Introduction in line 64 after the description "...a homologous HK-related module that, however, lacks key residues essential for HK function and is thus devoid of catalytic activity" (probably using references such as Yeh et al. PNAS, 1998, 95:13976-13981 and Shin et al. Nat Commun, 2016, 7:11545).

4. There are sentences that are likely incomplete throughout the manuscript. Thus, the authors need to revise the text more carefully. Here are some examples:

Line 22 in the Abstract, complexation? (maybe complex formation)

Lines 23-24, incomplete sentence "..., among them the phytochrome-interacting factors (PIF), a family of transcription factors". Need to revise the sentence, such as "..., among them the phytochrome-interacting factors (PIF), a family of transcription factors, have been known to play important roles in phytochrome-mediated signaling pathways."

Line 33, revise "..., and which formed.." (probably by removing "and")

Lines 56-57, incomplete sentence "... and population of the metastable, far-red-absorbing Pfr state", so need to revise. (probably, "... and population of the metastable, far-red-absorbing Pfr state increases")

Lines 81-82, need to revise the sentence, for example "..., which localize to the APA motifs (active phytochrome A binding) of these PIFs located in somewhat closer to C-terminal of the APB motifs"

Lines 100-108, the contents are overlapped with lines 319-325. The authors need to re-write the parts.

Lines 311-312, revise the sentence "In this study, we have dissected the light-regulated PPIs the AtPhyB PCM enters with the AtPIFs 3 and 6" [possibly, "In this study, we have dissected the light-regulated PPIs between AtPhyB PCM and AtPIF3/6"].

Lines 373-375, "Although the reliance on BV in this system is advantageous for applications in vivo, the degree of light regulation appears much lower than in the present setup based on plant Phys".

Please explain more specifically why the present system is better than bacterial systems with BV.

5. As for Table and Figures,

- In Fig. 1A, according to recent publications (such as Qiu et al., Nat Commun, 8:1905), it would be better to use PAS-A, PAS-B, and HKRD, instead of PAS, PAS, and HKR. In addition, please include NTE (N-terminal extension).

- Check "Px.LS" in Fig. 1E vs. P3.SL and P3.SL in Suppl. Table 1 / P3.SL, P6.SL and P1.SL in Suppl. Fig. 7 (Probably Px.LS should be used).

- Indicate what (colors) are Pfr and Pr forms of phyB in Fig. 2A for readers.

- There are no red lines in Suppl. Fig. 3, so revise the legend.

- Figure 4 legend, revise "Data points show mean n = 3 biological replicates" [maybe "Data points

show means of three biological replicates" etc.)

6. Update references: as examples, 51 (J Mol Biol. 2019; 431(17):3029-3045) and 54 (Annu Rev Biochem. 2015; 84:519-50). In addition, the reference formats should be consistent. Currently, both uppercase and lowercase letters are used for the titles of references.

Dear Dr. Dominique Morneau, dear Reviewers,

we are grateful for the careful study of our manuscript and the constructive remarks. We enclose a revised version of our manuscript which addresses these comments point-by-point as detailed below (original comments in italics, responses in red):

Reviewer #1:

(...)

Minor points:

1. If possible, could the authors include as supplemental figures images of SDS-PAGE gels showing the molecular weight, integrity and relative abundance of the E.coli expressed segments examined (and western blots of the mammalian expressed proteins, if a tag is present)?

We include as Suppl. Fig. 9 images of SDS-PAGE gels showing all proteins purified in this study. In the experiments in mammalian cells, the PIF proteins did not carry any tag, as it might interfere with system function and robustness. That said, previous studies had revealed that SEAP reporter expression strictly requires the functional expression of both *AtPhyB* and *AtPIF* (cf. Müller *et al. Nucleic Acids Res* (2013)). At a qualitative level, the observation of SEAP expression above background thus indicates functional expression of the *AtPIF* variant in question.

2. The manuscript could benefit from further highlighting/expanding the discussion on:

a) how these findings could be potentially utilised to enhance our understanding of the function of the native plant proteins (PIF-phyB)

b) specific examples of novel, optogenetic applications of the toolkit presented in this manuscript in addition to the ones stated.

We welcome this suggestion and have added to the manuscript several sentences on points a) and b).

Reviewer #2:

(...)

I am overall enthusiastic about its publication, but do have the following minor comments.

*Point 1. The light-regulated AtPhyB:AtPIF interaction offers a reversible regulation, the system can be switched ON and OFF. In Fig. 5, the authors tested the interaction between the AtPIF variants and AtPhyB PCM in a previously reported system (Müller *et al* 2014 *Nature Protocols* and Müller *et al* 2014 *Molecular BioSystems*), and the system has been shown to be reversible in previous reports. However results presented in Fig 5 did not provide data of the OFF switch. The dark and red-light conditions as described in line 508-509 were cells kept in two conditions respectively. How will the SEAP gene expression respond to a far-red light illumination after the red light illumination? Can the truncated variants of AtPIFs influence the far-red light caused OFF switch in vivo? Can the extremely higher SEAP expression caused by the Px.AA variants be reversed by far-red light? And also, how long does the red light illumination and darkness last? There is no “indicated period of time” (line 508) in Fig. 5 and supplementary Fig. 7. What is neg. control in Fig. 5?*

We thank the reviewer for bringing up this important point. To address it, we now performed experiments on cells first exposed to red light for 24 h, followed by exposure to far-red light for another 24 h. In these cells, SEAP reporter levels largely subsided to the levels observed in cells incubated in darkness for the entire 48 h. The data illustrate the photoreversibility of the system and are included as Suppl. Fig. 8.

We added the missing information on the incubation times and the negative control to the methods section and to the legend to Fig. 5.

Point 2. Fig. 6 used a similar system as Levskaya et al 2009 Nature. The 2009 report indicated the tandem PAS repeats of PhyB C-terminal OPM region is required for the reversible membrane recruitment of PIF6 in vivo. Since the AtPhyB PCM used in Fig 6 does not contain the tandem PAS repeats and the results shown went through a far-red light illumination followed by a red light illumination, I am wondering can the membrane recruitment of AtPIF6 variants be reversed by another far-red light illumination after the red light. And also what is the intensity and duration of the red and far-red light used in Fig. 6 results?

Although the early paper by Levskaya *et al.* indeed reported irreversibility of interaction for the isolated PCM, several later studies obtained photoreversible PIF interactions with the PCM. Examples include Buckley *et al. Dev Cell* 2016 and Yousefi *et al. eLife* 2019, and we have added these references and a remark to the text. In any case, we appreciate this comment and included experiments that show the reversibility of membrane recruitment between the AtPhyB PCM (residues 1-650) and the newly developed AtPIF6 variants (see revised fig. 6).

Moreover, we included information on the employed light intensities and their determination. Notably, the intensities used presently are similar to those used in a previous study on the AtPhyB:AtPIF3/6 system by Pathak *et al. (ACS Synth Biol* 2014) which we now cite in the manuscript.

Point 3. Multiple evidence in the current manuscript seems to show the linker region between APB.A and APB.B or part of the linker region are important for the AtPhyB PCM and AtPIF3/6 or AtPIF1 variants interaction: supplementary Fig. 4 D, F, H and I; the Pfr state KD values for P3.As and P6.As (compared to the Pfr state KD values for P3.A and P6.A respectively) in Table 1 and the Pfr state KD values for P6. fus (compared to the Pfr state KD values for P6) in Table 1; P6.As compared to P6.A in Fig. 5B, P3.As compared P3.A in Fig. 5C and P1.As compared to P1.A in Fig. 5D. Do the authors have any explanation about this?

We concur with this reviewer's astute observation. As we duly reported in our original manuscript, the PIF linker region apparently affects the affinity, if mostly to comparatively limited extent (see Table 1). We conclude that the linker can contribute to binding, which is arguably best seen for the K_D values of PxA vs. PxA.s (see Table 1).

In most cases, this effect is moderate at most; as an exception, the binding affinity takes a larger hit when the entire linker is removed from P6, thus yielding P6.fus with an around 20-fold higher K_D . In this specific case, we speculate that the reduced spacing impairs the productive interplay between the APB.A and APB.B motifs of AtPIF6. Hence, the affinity for the P6.fus variant is reduced to about the level of P6.A that entirely lacks APB.B.

Point 4. The names of P3.P1L, P6.P1L, P3.SL and P6.SL in supplementary Table 1 and supplementary Fig.7 are not consistent with the main text and Fig. 1E.

We corrected this mistake.

Reviewer #3:

(...)

Overall, the novelty of this study is not strong, because optogenetics studies such as light-regulated gene expression and photoreversible recruitment to plasma membrane are already reported (see the references in the manuscript) and the APB sequence of PIF6 consisting of 22 amino acids is already known to be sufficient for the interaction with AtphyB (see reference 38). However, as the authors mentioned, systemic studies of PPIs between AtphyB and PIFs are not reported, so the results of this manuscript might be useful for the future study of phytochrome-PIFs interactions and also optogenetics study especially with plant phytochromes.

The contents and organization of the manuscript are easy to follow the work done for this study. However, here are some comments to improve the manuscript.

We respectfully beg to differ regarding the claim of limited novelty. As also recognized by reviewers #1 and #2, our work reports fundamental, previously unavailable and quantitative insight into the Phy:PIF PPI. These findings transcend the previous knowledge and are directly relevant for both optogenetics and plant signal transduction. We appreciate that this reviewer concurs that systematic data on this PPI have so far been lacking, and that hence our results stand to be useful for future studies.

1. There are no data to see the quality of proteins used in this study. Thus, it would be better to include SDS-PAGE gels to show purified proteins used in this study. In addition, as the authors mentioned that the C-terminal output module (OPM) contributes to light-regulated PPIs and is possibly important for eliciting physiological responses, is there any result of PPIs between full-length or OPM of AtphyB and the APB variants?

We included as Suppl. Fig. 9 images of SDS-PAGE gels of all proteins purified in this study (see response to reviewer #1).

In our manuscript, we concentrated on mapping the interaction between the AtPhyB PCM and the AtPIF variants. We appreciate this reviewer comment and plan to study interactions of full-length AtPhyB (i.e. including the OPM) and the AtPIF proteins in the future.

In Suppl. Fig. 7, P3.SL and P1.SL showed light-dependent expression but P6.SL did not. Is there any explanation for this result?

We repeated this experiment several times with the same outcome, i.e. light-dependent expression observable for P1.LS and P3.LS but not for P6.LS. Although it is challenging to rationalize these findings at the molecular level, we note similar trends in the Px.LP1 variants (bearing the linker segment derived from AtPIF1): the variant P6.LP1 exhibited noticeably lower overall SEAP expression than the corresponding variant P3.LP1. For as-of-yet unclear reasons, P6 may hence tolerate foreign insertions (such as the synthetic and the P1-derived linkers) less well than P3.

2. It would be recommended to use “phyB”, instead of “PhyB”, for phytochrome B, according to the nomenclature of phytochromes (*Plant Cell*, 1994, 6:468–471). Similarly, use “phy” instead of “Phy” for phytochrome, and AtphyB and AtphyA instead of AtPhyB and AtPhyA etc.

We appreciate the reference to the 1994 *Plant Cell* paper which advocated common denominations for phytochrome genes, alleles and proteins, thereby reconciling previously

used, sometimes conflicting names. Specifically, the denomination ‘phy B’ has been suggested for the phytochrome B holoprotein.

However, a survey of the recent literature reveals that several naming schemes are in use. In the area of optogenetics, it is indeed the abbreviation ‘PhyB’ that prevails, see e.g., Levskaya *et al. Nature* 2009 and Müller *et al. Nucl Acids Res* 2013. Moreover, we have used this abbreviation in our previous publications, and for sake of consistency, we thus kept the naming scheme ‘AtPhyB’ etc.

3. It might be better to mention about the serine/threonine kinase activity observed in plant phytochromes in the Introduction in line 64 after the description “...a homologous HK-related module that, however, lacks key residues essential for HK function and is thus devoid of catalytic activity” (probably using references such as Yeh *et al. PNAS*, 1998, 95:13976-13981 and Shin *et al. Nat Commun*, 2016, 7:11545).

We added a sentence and citations of these references.

4. There are sentences that are likely incomplete throughout the manuscript. Thus, the authors need to revise the text more carefully. Here are some examples:

We have carefully studied the sentences that this reviewer objects to. Although we maintain that the sentences are by and large correct, we acknowledge that tastes differ regarding sentence structure and wording. We have hence addressed these comments as follows.

Line 22 in the Abstract, *complexation?* (maybe complex formation)

Changed to ‘complex formation’. (Note, however, that the word ‘complexation’ exists and indeed means ‘complex formation’.)

Lines 23-24, *incomplete sentence “..., among them the phytochrome-interacting factors (PIF), a family of transcription factors”*. Need to revise the sentence, such as “..., among them the phytochrome-interacting factors (PIF), a family of transcription factors, have been known to play important roles in phytochrome-mediated signaling pathways.”

This sentence is correct and complete. Not changed.

Line 33, revise “..., and which formed..” (probably by removing “and”)

We reworded and thus clarified the sentence.

Lines 56-57, *incomplete sentence “... and population of the metastable, far-red-absorbing Pfr state”, so need to revise. (probably, “... and population of the metastable, far-red-absorbing Pfr state increases”)*

The sentence ‘absorption of red light triggers bilin isomerization (...) and population (...)’ is correct. Not changed.

Lines 81-82, *need to revise the sentence, for example “..., which localize to the APA motifs (active phytochrome A binding) of these PIFs located in somewhat closer to C-terminal of the APB motifs”*

We inserted a comma to clarify this sentence.

Lines 100-108, *the contents are overlapped with lines 319-325. The authors need to re-write the parts.*

Lines 100-108 refer to the last paragraph of the introduction which summarizes the main findings of our manuscript. We deem it equally common and sensible to include these

sentences in the introduction. Crucially, the author guidelines for *Comm Biol* also require this summary paragraph at the end of the Introduction. A certain redundancy with the Discussion section (lines 319-325) therefore appears inevitable. For reasons of legibility and clarity, we hence left the text as is.

Lines 311-312, revise the sentence “In this study, we have dissected the light-regulated PPIs the AtPhyB PCM enters with the AtPIFs 3 and 6” [possibly, “In this study, we have dissected the light-regulated PPIs between AtPhyB PCM and AtPIF3/6”].

We changed this sentence.

Lines 373-375, “Although the reliance on BV in this system is advantageous for applications in vivo, the degree of light regulation appears much lower than in the present setup based on plant Phys”. Please explain more specifically why the present system is better than bacterial systems with BV.

We respectfully note that we explicitly mentioned that the BV usage is an advantage of the bacterial systems. By contrast, we did not intend to convey that the plant-based system was generally better, nor did we make this claim. That said, we toned down the statement by deleting the second part of the sentence and now merely note that the reliance on BV is advantageous *in vivo*.

5. As for Table and Figures,

- In Fig. 1A, according to recent publications (such as Qiu et al., Nat Commun, 8:1905), it would be better to use PAS-A, PAS-B, and HKRD, instead of PAS, PAS, and HKR. In addition, please include NTE (N-terminal extension).

Done.

- Check “Px.LS” in Fig. 1E vs. P3.SL and P3.SL in Suppl. Table 1 / P3.SL, P6.SL and P1.SL in Suppl. Fig. 7 (Probably Px.LS should be used).

We thank the reviewer for spotting this error which we have now corrected.

- Indicate what (colors) are Pfr and Pr forms of phyB in Fig. 2A for readers.

Done. We also put a corresponding explanation in the legend to Fig. 1C.

- There are no red lines in Suppl. Fig. 3, so revise the legend.

Done.

- Figure 4 legend, revise “Data points show mean n = 3 biological replicates” [maybe “Data points show means of three biological replicates” etc.]

We corrected this sentence.

6. Update references: as examples, 51 (J Mol Biol. 2019; 431(17):3029-3045) and 54 (Annu Rev Biochem. 2015; 84:519-50). In addition, the reference formats should be consistent. Currently, both uppercase and lowercase letters are used for the titles of references.

We thank the reviewer for the attentive reading. We corrected references 51 and 54. We have capitalized words in the titles of the references as done in the original articles. Hence, there are inevitably inconsistencies here.

Beyond the specific reviewer requests, we also edited the manuscript such that it conforms to the guidelines of *Communications Biology*. We consider the revised manuscript much improved and hope that it is now suitable for publication in *Communications Biology*. Thank you for your consideration and best regards,

Andreas Möglich

REVIEWERS' COMMENTS:

Reviewer #1 (Remarks to the Author):

The authors have addressed all my comments.

Reviewer #2 (Remarks to the Author):

The authors have addressed all my concerns in this revised manuscript and the revised version has been improved. I am enthusiastic about its publication.

Reviewer #3 (Remarks to the Author):

The authors (Golonka et al.) have resubmitted this manuscript and included a detailed response to the previous reviews, such as SDS-PAGE gels of purified proteins (Suppl. Fig. 9), revision of Fig. 1A, and other corrections in sentences and figure legends. Thus, I think the authors have addressed most of my previous concerns.

There is only a minor comment:

Reference 13 would be "Nat. Commun. 7, 11545 (2016)", instead of "Nat. Commun. 7, 1–13 (2016)".